# Latent Equivariant Operators for Robust Object Recognition: Promises and Challenges

**Minh Dinh**
Department of Computer Science
Dartmouth College
Hanover, NH 03755, USA
Minh.T.Dinh.GR@dartmouth.edu

**Stéphane Deny**
Departments of Computer Science &
Neuroscience and Biomedical Engineering
Aalto University, Espoo, Finland
stephane.deny.pro@gmail.com

## Abstract

Despite the successes of deep learning in computer vision, difficulties persist in recognizing objects that have undergone group-symmetric transformations rarely seen during training—for example objects seen in unusual poses, scales, positions, or combinations thereof. Equivariant neural networks are a solution to the problem of generalizing across symmetric transformations, but require knowledge of transformations *a priori*. An alternative family of architectures proposes to *learn equivariant operators* in a latent space, from *examples* of symmetric transformations. Here, using simple datasets of rotated and translated noisy MNIST, we illustrate how such architectures can successfully be harnessed for out-of-distribution classification, thus overcoming the limitations of both traditional and equivariant networks. While conceptually enticing, we discuss challenges ahead on the path of scaling these architectures to more complex datasets. Our code is available at https://github.com/BRAIN-Aalto/equivariant_operator.

## 1 Introduction

Deep networks have progressed to match or even outperform humans on many image recognition benchmarks (He et al., 2015; Vasudevan et al., 2022; Dehghani et al., 2023). However, deep networks mostly excel on testing sets which are *identically distributed* (iid) to the training set. This high performance on iid benchmarks is not necessarily indicative of their performance in scenarios that have rarely been visited during training, in the so-called *out-of-distribution* domain. For example, state-of-the-art deep networks have shown to be brittle on tasks of recognizing objects in unusual poses, scales, or positions (Alcorn et al., 2019; Madan et al., 2021; Ibrahim et al., 2022; Madan et al., 2022; Abbas & Deny, 2023; Ollikka et al., 2025). Such scenarios can often be described in the language of *group theory*: changes of pose, scale, and position are indeed the result of *group transformations* acting on visual objects.

Several approaches have been proposed to increase robustness of deep networks to group transformations that are absent or only partially present in the training dataset. First, **equivariant neural networks** offer guarantees of robustness to target group transformations (Cohen et al., 2019; Bekkers, 2019). However, such approaches require complete *a priori* knowledge of the transformation. In particular, the transformation group structure (e.g., cyclic group of a certain order) and its specific representation (e.g., rotations or translations) must be specified mathematically to construct the relevant architecture. Second, **data augmentations schemes**, used in combination with either supervised or self-supervised learning objectives (e.g., Benton et al., 2020; Zbontar et al., 2021; Brehmer et al., 2024), can ensure some level of invariance to pre-specified transformations. However, for optimal results, transformations need to be sampled uniformly across the entire range of parameters seen at test time (Gerken & Kessel, 2024; Perin & Deny, 2025). This is not always possible, in particular when one is only given *examples* of transformations *in a limited range*. In a third line of work, methods have been proposed that learn group transformations from examples, hereafter referred to as **latent equivariant operator** methods (Culpepper & Olshausen, 2009; Sohl-Dickstein et al., 2017; Connor & Rozell, 2020; Dupont et al., 2020; Bouchacourt et al., 2021; Keller & Welling, 2021; LeCun, 2022; Connor et al., 2024). These methods learn an encoder jointly with a latent space operator, such that after training the latent space operator produces transformations that

are approximately equivariant to the transformations present in the dataset. These methods offer a promising alternative for out-of-distribution object recognition.[1]

**Here, we illustrate how equivariant latent operator methods can successfully be applied to out-of-distribution classification problems.** For simplicity, we strip down the complexity of the datasets and models to their minimal components necessary to establish the superiority of these methods over others. On rotated and translated noisy MNIST, we train a neural encoder jointly with a latent operator to learn equivariance over a limited range of transformations. At the same time, we train a classifier taking as input the latent space, on the same range of transformations. At inference time, we test our model over a range of transformations that were not seen during training, assessing its extrapolation and composition abilities. We use a K-nearest-neighbour strategy to select the operator action most likely to revert the object back to its canonical pose. We establish the superiority of our method over comparable architectures on classifying input samples outside the training range. Although our work builds on previous work—e.g., Bouchacourt et al. (2021); Connor & Rozell (2020)—we extend this prior work in several ways: (1) we demonstrate that latent equivariant operator methods can be used for classification *outside* the range of transformations seen during training, (2) while not specifying transformation parameters at test time, (3) and with a learnable operator necessitating only the specification of a weak periodicity prior. We conclude with a discussion on the challenges ahead on the path of scaling these methods to real world images and more complex datasets.

## 2 METHODS

**Dataset**  We present experiments on a task of MNIST digit classification (LeCun et al., 2010). We extract the digit by thresholding pixel intensities above 128, recolor the digit in blue, and place it onto a random black-and-white checkerboard background, acting as noise to be ignored by the classifier. The digit is transformed by either a rotation or X-Y translations. Rotations are discretized in steps of $36°$, yielding 10 distinct elements. Translations use a stride of 2 pixels along each axis on the $28 \times 28$ grid (with periodic boundary condition), resulting in a translation group of order 14 per axis. Variants of the same digit share the same noise background. To avoid confusion with class '6' when rotating, we exclude class '9' from the dataset.

**Architecture**  We use a simple feed-forward architecture. The encoder consists of a single linear layer that maps the flattened input into a latent representation to be transformed with the shift operator. When there are a combination of transformations, we use stacked encoders and operators for each transformation. We set the latent dimensionality to 70 to accommodate the orders of the transformation groups considered. The **pre-defined** latent operator follows the discrete construction of Bouchacourt et al. (2021): each block is a shift matrix with a size equal to the order of the corresponding transformation group and is repeated along the diagonal to match the latent dimensionality. To evaluate whether equivariance can be learned from scratch, we also consider a **learnable** operator variant initialized as the orthogonal factor $Q$ of a QR decomposition[2] of a randomly sampled matrix. A classifier, stacked after the encoder, is a two-layer MLP with a sigmoid nonlinearity in the hidden layer, producing class logits from the transformed latent features. Further details are provided in Appendix A.

**Training**  Given a training sample $(x, y)$, we generate two views by applying discrete transformations parameterized by $k_1$ and $k_2$: $x_1 = T^{k_1}(x), x_2 = T^{k_2}(x)$, where $T$ denotes a group transformation family and $k_1, k_2$ are sampled transformation parameters. As illustrated in Figure 1, each transformed view is first mapped to a canonical representation using the corresponding inverse shift operator. Specifically, the two views' canonicalized embeddings are obtained as

$$Z_1 = \varphi^{-k_1} f_E(x_1), \quad Z_2 = \varphi^{-k_2} f_E(x_2).$$

---

[1]Latent equivariant operator methods should be distinguished from disentanglement methods; the latter can be seen as special cases of the former, where the latent operator is confined to a subspace (Higgins et al., 2018)—but see topological defects induced by subspace disentanglement in Bouchacourt et al. (2021).

[2]QR initialization offers a stable orthogonal starting point for optimization; simpler initializations such as zero or unconstrained random matrices led to less reliable convergence in our experiments.

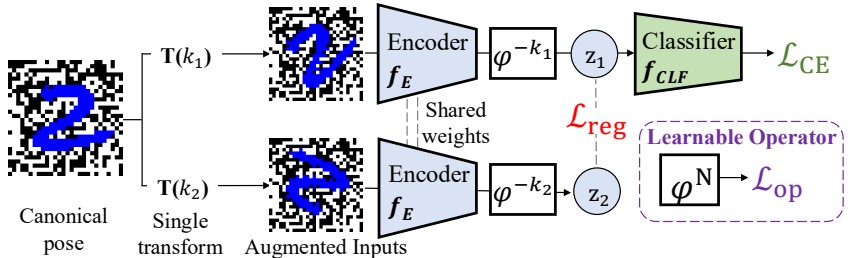

Figure 1: **Pipeline for handling single transformation.** Two transformed views of the same input are encoded by a shared encoder $f_E$ and mapped to a canonical representation using inverse shift operators $\varphi^{-k_1}$ and $\varphi^{-k_2}$, yielding embeddings $Z_1$ and $Z_2$. The embedding $Z_1$ is used for classification via a MLP $f_{\text{CLF}}$ optimized with the cross-entropy loss $\mathcal{L}_{\text{CE}}$, while a representation consistency loss $\mathcal{L}_{\text{reg}}$ encourages alignment between $Z_1$ and $Z_2$ that both correspond to the canonical pose. When the operator is learned, we add an extra term $\mathcal{L}_{\text{op}}$ to the loss.

We encourage consistency between the learned representations by minimizing the distance between these canonical embeddings:

$$\mathcal{L}_{\text{reg}} = \|Z_1 - Z_2\|_2^2. \tag{1}$$

The canonicalized representation of the first view is fed to the classifier head:

$$\mathcal{L}_{\text{CE}} = \text{CrossEntropy}(f_D(Z_1), y).$$

The final training objective is

$$\mathcal{L} = \mathcal{L}_{\text{CE}} + \lambda \mathcal{L}_{\text{reg}}, \tag{2}$$

where $\lambda$ controls the strength of the regularization. When using a learnable operator, we aim to preserve the periodic properties of the operator by adding to the objective in Equation 2 another term that encourages the periodicity of the operator group:

$$\mathcal{L}_{\text{op}} = \|\varphi^N - I\|_2, \tag{3}$$

where $N$ is the order (number of unique elements) of the group represented by the operator. Throughout all experiments, the order of our learnable operator was set to 70, equal to the latent dimensionality, which is substantially larger and did not need to be tailored to the true period of the underlying transformation (e.g., 10 for rotations or 7 for translations).

**Inference**    In the absence of explicit transformation labels at inference time, we infer the pose of an input via a $K$-nearest neighbor (k-NN) search over a reference set of canonical embeddings. We first construct a class-agnostic reference database by collecting $N$ validation samples $x_j$ with known transformation indices $\ell_j$ and mapping them back to a canonical pose using the inverse operator:

$$\mathcal{R} = \left\{ r_j = \varphi^{-\ell_j} f(x_j) \right\}_{j=1}^N.$$

Given a test input $x$, we evaluate its embedding under each candidate transformation operator $\{\varphi_\ell\}$:

$$z_\ell = f(\varphi_\ell(x)), \quad \ell \in \mathcal{G},$$

where $\mathcal{G}$ denotes the set of discrete transformation indices. We compute Euclidean distances between each transformed embedding $z_\ell$ and all reference embeddings $r_j \in \mathcal{R}$. The predicted transformation index is obtained by majority voting over the indices associated with the $K$ nearest reference matches:

$$\hat{\ell} = \text{mode}\big(\text{Top}K\big(\{\|z_\ell - r_j\|_2\}_{\ell,j}\big)\big), \tag{4}$$

where $\text{Top}K(\cdot)$ returns the transformation indices corresponding to the $K$ smallest distances among all reference comparisons. The embedding $z_{\hat{\ell}}$ is then selected and passed to the classifier.

## 3   RESULTS

All implementation details are provided in Appendix B.

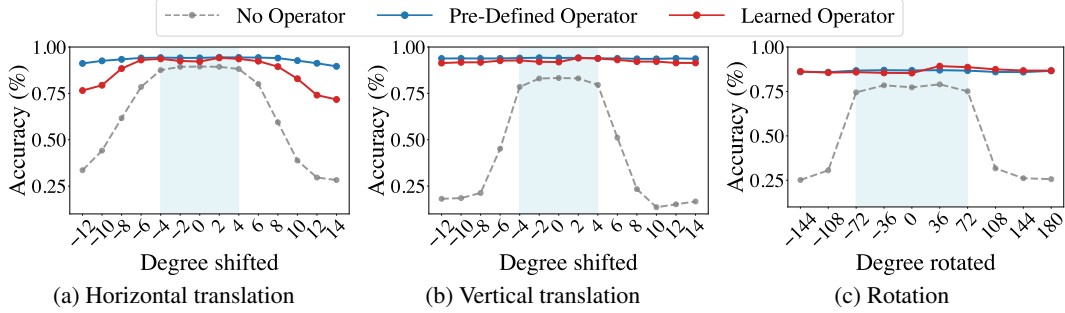

Figure 2: **Classification accuracy as a function of transformations on MNIST.** The shaded region denotes the range of translations observed during training.

**Performance on unseen degrees of a single transformation**  We designate a subset of transformations as in-domain during training. For rotations, we use angles of $\{-72° - 36°, 0°, 36°, 72°\}$. For translations, we use shifts by $\{-4, -2, 0, 2, 4\}$ pixels. Figure 2 illustrates classification accuracy as a function of horizontal translation, vertical translation, and rotation. For the baseline model—a model with the same encoder-classifier architecture but in which no operator was applied during training or inference—accuracy peaks within the training range and decreases sharply as inputs move further into unseen transformations, forming a pronounced bell-shaped curve. In contrast, models equipped with latent operators exhibit nearly flat accuracy profiles across the entire transformation range, indicating stable extrapolation. The learned-operator variants show similar trends, with slightly increased variance in accuracy across degrees of transformation, suggesting that equivariant structure can be recovered even when the operator is learned rather than fixed. We report the exact numerical results in Appendix C.1, and an ablation on the k-NN method in Appendix C.2.

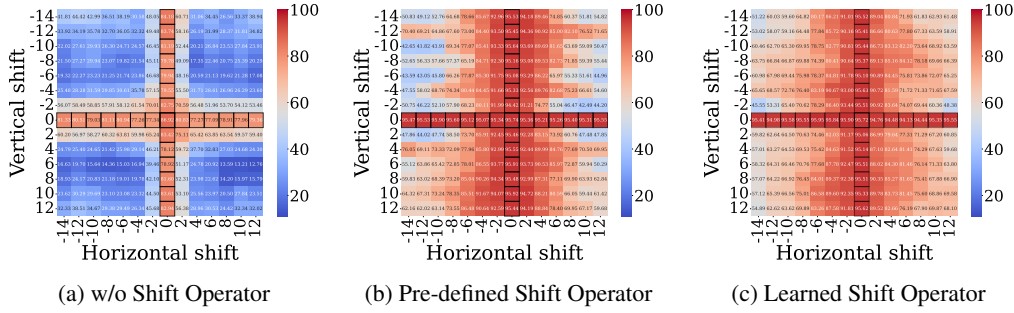

Figure 3: **Test accuracy heatmaps under joint horizontal (rows) and vertical (columns) translations.** The bordered cross indicates transformations observed during training.

**Performance on unseen combination of transformations**  Figure 3 shows classification accuracy under joint horizontal and vertical translations, using the stacked operator version of our model. When no shift operator is used (Figure 3a), accuracy deteriorates rapidly outside the training region, indicating poor generalization to unseen translation combinations. In contrast, both the pre-defined (Figure 3b) and learned (Figure 3c) shift operators generalize well beyond the training cross, maintaining high accuracy across most unseen combinations. Notably, the learned shift operator attains comparable, and in some corner regions slightly higher accuracy than the pre-defined operator, suggesting that data-driven operator learning can recover effective equivariant structure without explicit specification.

## 4 DISCUSSION

Here we demonstrated, in a minimal setup, that latent equivariant operators can be leveraged to classify out-of-distribution samples—by means of extrapolation and combination of symmetric transformations. Unlike in equivariant neural networks, the symmetries did not need to be mathemat-

ically pre-specified. Unlike in data augmentation schemes, the method did not require coverage of the full range of transformation parameters during training. Intuitively, this is because the recursive application of operators in latent space allows the extrapolation of transformations beyond the training range, and in unseen combinations. Furthermore, the method does not require knowledge of the transformation parameter at inference time, as the latent transformation can be estimated through canonicalization in representation space. **Latent equivariant operator methods thus offer a promising avenue for robust, human-like[3] object recognition, and warrant further investigation.**

To date, latent equivariant operator methods have not been shown to scale to large and complex datasets in the out-of-distribution domain. In this work, our experiments use a minimal controlled setup (MNIST with synthetic noisy backgrounds) and focus on invariant classification. Evaluating these methods on more realistic noise sources, broader transformation families, and more complex datasets, therefore, remains an important direction for future work. **At the same time, several theoretical and implementation challenges remain.** First, we do not know theoretically the certainty with which we can expect operators to remain equivariant beyond the training range of transformation parameters. Empirically, we find classification performance to degrade somewhat outside the training range. Second, we do not know at what layer of an architecture such operators should be situated. For the affine transformations explored in this minimal study, a single linear layer suffices to recast the representation in a way suitable to the operator (whether learned or fixed), as described in previous theory (Bouchacourt et al., 2021). For more complex transformations, for example transformations happening in a latent space not directly accessible from pixel space (e.g., in-depth 3D rotation), it is unclear how many layers would be required. Empirically, latent equivariant operators have been shown to successfully emulate in-depth 3D rotations (Dupont et al., 2020). Finally, the choice of the functional form of latent operators is another open question, involving fundamental notions in *representation theory* and *topology*. The resolution of these theoretical questions should provide guidance for the successful implementation of latent equivariant operator approaches at scale.

## ACKNOWLEDGMENTS

We thank members of the BRAIN lab at Aalto University, Ivan Vujaklija, Mansour Taleshi for useful discussions, and Fabio Anselmi and T. Andy Keller for helpful comments on the manuscript. This work was supported in part by the Academy of Finland under Grant 3357590 for S.D, and by the Helsinki Institute for Information Technology (HIIT) under Contract No. 9125064HIIT for M.D.T.

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

# A METHODOLOGY DETAILS

## A.1 PRELIMINARY ON SHIFT OPERATOR

**Model components.** We decompose the model into an encoder $f_E : \mathcal{X} \to \mathcal{Z}$, which maps an input $x \in \mathcal{X}$ to a latent representation, and a classifier $f_{\text{CLF}} : \mathcal{Z} \to \mathcal{Y}$, which predicts the output label from the representation. Unless otherwise stated, all encoders are linear in this study.

We consider a family of discrete affine transformations that form a cyclic group, such as translations or rotations. Our goal is to learn an equivariant representation with respect to this group action.

Formally, let $T^k$ denote a transformation applied in the input space, parameterized by $k$, and let $\hat{T}^k$ denote the corresponding transformation acting on the representation space. We seek an encoder $f_E$ such that

$$f_E(T^k x) = \hat{T}^k f_E(x), \tag{5}$$

or equivalently,

$$T^k x = f_E^{-1}\left(\hat{T}^k f_E(x)\right), \tag{6}$$

where $f_E^{-1}$ denotes a (possibly implicit) decoder. Although $T^k$ and $\hat{T}^k$ are parameterized by the same group element, they need not share the same functional form.

One approach to achieving equivariance is to explicitly map representations to a canonical pose, thereby factoring out the effect of the transformation. This perspective motivates the use of a *shift operator* Bouchacourt et al., 2021, which models the group action in the representation space. Concretely, the shift operator, which has the same dimensionality as the latent embedding, is constructed in Kronecker product form using the building block $M$:

$$M^k := \begin{bmatrix} 0 & 0 & \cdots & 1 \\ 1 & 0 & \cdots & 0 \\ 0 & 1 & 0 & \cdots \\ \vdots & & \ddots & \vdots \\ 0 & \cdots & 1 & 0 \end{bmatrix}^k \tag{7}$$

The matrix $M$ corresponds to the elementary generator of the transformation group and has size equal to the order of the group. For a shift of degree $n$, the corresponding shift operator is obtained by applying the elementary shift $n$ times, which is equivalent to using the matrix power $M^n$ as the building block along the diagonal.

Under this formulation, sequential transformations compose additively in the representation space. In particular, for two transformations $T^{k_1}$ and $T^{k_2}$, we have

$$T^{k_2} T^{k_1} x = f_E^{-1}\left(M^{k_2} M^{k_1} f_E(x)\right) = f_E^{-1}\left(M^{k_1+k_2} f_E(x)\right), \tag{8}$$

which reflects the underlying group structure.

Importantly, the shift operator formulation does not require explicit knowledge of the transformation parameter applied to each input. Instead, it only assumes knowledge of the cycle order of the transformation group, making it well suited to scenarios with limited or missing transformation supervision.

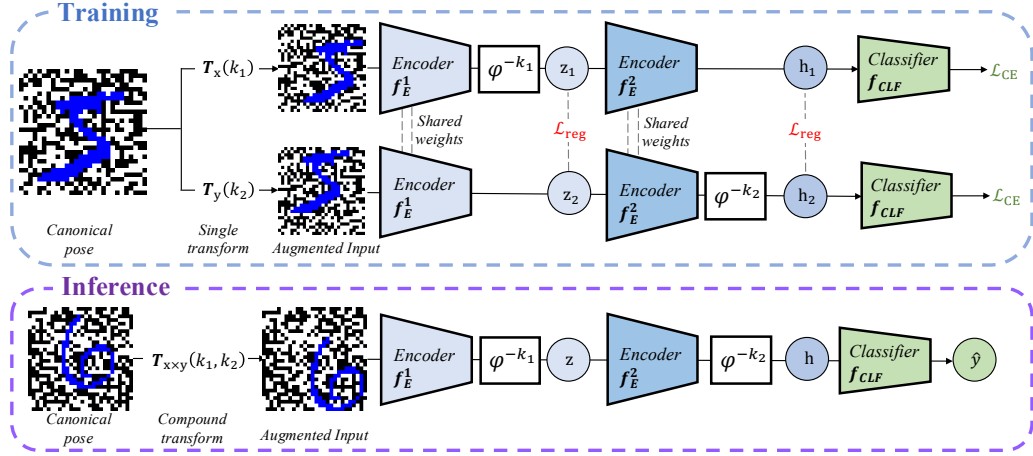

Figure 4: **Pipeline of operator-based classification under compound transformations.** Top (Training): A canonical input is transformed along individual axes to generate augmented views. Shared encoders map each view into a representation space, where inverse operators $\varphi^{-k}$ align embeddings back to a canonical pose. Bottom (Inference): Given an input undergoing a compound transformation $T_{x,y}(k_1, k_2)$, the model encodes the input and applies the corresponding inverse operators to recover a canonical representation for classification.

## A.2 EXTRAPOLATION WITH SHIFT OPERATOR

Extrapolation refers to making predictions on regions of the input domain that lie outside the range observed during training, in contrast to interpolation, which concerns predictions within the training domain. To study extrapolation under controlled conditions, we restrict the range of shift transformations observed during training in two ways.

Firstly, Section A.2.1 discusses transformations along a single degree of freedom (e.g., rotation or translation along one axis). In this case, although the full transformation group contains $N$ discrete degrees, we train the model using only a subset of $N_{\text{train}} < N$ transformation degrees and evaluate on the remaining unseen degrees.

Secondly, Section A.2.1 elaborates on transformations along multiple dimensions (*compound transformation*), for which we restrict the set of transformation pairs to a strict subset of the full Cartesian product: $\mathcal{K}_{\text{train}} \subset \mathcal{K}_x \times \mathcal{K}_y$. Evaluation is then performed on unseen compositions $(\mathcal{K}_x \times \mathcal{K}_y) \setminus \mathcal{K}_{\text{train}}$, which tests the model's ability to generalize to novel combinations of transformations.

### A.2.1 SINGLE TRANSFORMATION

We first study extrapolation under a single transformation dimension. During training, the model is exposed to only a restricted subset of transformation degrees, while evaluation is performed on unseen degrees outside this range. This setting directly tests whether the learned operator captures the underlying transformation structure rather than merely interpolating within the observed range.

From the perspective of group theory, many transformation families form a cyclic group whose elements are generated by repeated application of an elementary operator. Because group actions are closed under composition, transformations outside the training range can be expressed as compositions of transformations seen during training. Consequently, if the model learns the correct group action for a subset of elements, it can extrapolate to unseen transformations via recursive application of the same operator.

### A.2.2 COMPOUND TRANSFORMATION

In many practical settings, inputs undergo transformations that are compositional rather than separate, such as simultaneous translations along multiple axes. We refer to such cases as *compound*

*transformations*, where the overall transformation can be expressed as the composition of multiple elementary operators.

A direct approach would be to learn a distinct operator for every possible combination of transformations. However, this quickly becomes impractical: if there are $M$ transformation types, each with $N$ discrete variants, exhaustively observing all compound transformations requires $O(N^M)$ transformation instances. This not only leads to a combinatorial explosion in the number of operators, but also substantially increases the dimensionality of the latent representation.

Compound transformations can instead be decomposed into elementary components. Prior latent operator approaches Bouchacourt et al. (2021) employ stacked shift operators to sequentially undo each transformation, ensuring that the number of operators scales with the order of each elementary group. However, the method proposed in Bouchacourt et al. (2021) was trained on all possible compound poses, which incurs the same quadratic data requirement: observing all compositions requires $O(N^M)$ samples.

In contrast, our approach trains the model using only single-axis transformations, as denoted in Figure 4. During training, we expose the model only to individually transformed views along each axis. For a training sample x and two transformation $T_x, T_y$, we generate

$$x_x = T_x^{k_1}(x), \qquad x_y = T_y^{k_2}(x),$$

and obtain their canonicalized embeddings by applying the corresponding inverse operators:

$$Z_x = \varphi_x^{-k_1} f_E(x_x), \qquad Z_y = \varphi_y^{-k_2} f_E(x_y).$$

Consistency is enforced between these canonical representations using the same regularization objective as in the single-transformation case:

$$\mathcal{L}_{\text{reg}} = \|Z_x - Z_y\|_2^2, \tag{9}$$

while classification is performed on the canonicalized embedding.

This encourages the encoder to learn representations that are aligned under each elementary transformation. Although this separate transformation was technically compatible with Bouchacourt et al. (2021), the feasibility of learning compound transformations via exposure to single transformations during training exclusively was not explored.

At inference time, when the input is subject to a known compound transformation, similar to (Bouchacourt et al., 2021), we sequentially apply the corresponding inverse operators to recover a canonical representation before classification. By exploiting the compositional reuse of operator blocks, our method reduces both the size of the operator space and the number of required transformation observations from $O(N^M)$ to $O(NM)$, while handling compound transformations without introducing additional parameters or retraining.

### A.3  LEARNED OPERATOR

The hard-coded constructions presented in Section A.1 demonstrate the existence of a valid operator that satisfies the desired transformation behavior for this task. While these fixed operators provide a constructive proof of feasibility, they are not necessarily optimal when integrated into a larger learning pipeline. To allow the operator to adapt to the data and interact with other learned components, we parameterize the operator and optimize it jointly with the rest of the model. In practice, the operator is initialized using the orthogonal factor of a QR decomposition of a random matrix to provide a stable starting point for optimization.

In this setting, the true order of the transformation group may be unknown. We therefore fix the size of the operator building block to match the latent dimensionality, which serves as an upper bound on the transformation order. This design allows the learned operator to represent transformations of varying effective orders within a fixed-dimensional latent space.

## B  IMPLEMENTATION DETAILS

We use the standard train/test split of the MNIST dataset (LeCun et al., 2010), further partitioning the training set into training and validation subsets with an 80/20 split. For background generation,

Table 1: **Classification accuracy (%) under translation extrapolation.** Results for horizontal (x-axis) and vertical (y-axis) shifts. "Degree given" indicates whether the true translation degree is provided at test time. Blue-shaded columns denote degrees seen during training; unshaded columns are unseen.

| Operator | Degree given | k | -12 | -10 | -8 | -6 | -4 | -2 | 0 | 2 | 4 | 6 | 8 | 10 | 12 | 14 |
|---|---|---|---|---|---|---|---|---|---|---|---|---|---|---|---|---|
| **Translation on y-axis** | | | | | | | | | | | | | | | | |
| × | – | – | 18.151 | 18.519 | 21.299 | 45.145 | 78.490 | 83.016 | 83.294 | 83.072 | 79.613 | 51.129 | 23.357 | 13.602 | 15.182 | 16.717 |
| Fixed | ✓ | – | 95.885 | 95.985 | 95.996 | 95.952 | 96.018 | 96.185 | 96.063 | 96.074 | 95.751 | 96.018 | 95.807 | 95.707 | 95.629 | 95.707 |
| Fixed | × | 1 | 93.816 | 93.872 | 93.816 | 93.849 | 94.061 | 94.216 | 94.094 | 94.183 | 93.894 | 93.861 | 93.638 | 93.616 | 93.872 | 93.683 |
| Fixed | × | 3 | 93.594 | 93.849 | 93.938 | 93.861 | 94.150 | 94.261 | 94.606 | 93.894 | 93.282 | 93.582 | 93.427 | 93.638 | 93.538 | 93.293 |
| Fixed | × | 10 | 93.605 | 93.727 | 93.916 | 93.916 | 94.061 | 94.194 | 94.161 | 94.372 | 93.816 | 93.705 | 93.516 | 93.571 | 93.438 | 93.582 |
| Learned | ✓ | – | 94.628 | 95.329 | 95.429 | 96.185 | 96.274 | 95.840 | 95.985 | 96.040 | 96.263 | 96.118 | 95.373 | 95.106 | 94.951 | 94.773 |
| Learned | × | 1 | 91.347 | 91.736 | 91.758 | 92.626 | 92.782 | 92.025 | 91.914 | 94.061 | 93.838 | 93.082 | 92.137 | 92.148 | 91.425 | 91.425 |
| Learned | × | 3 | 74.964 | 78.167 | 87.543 | 92.726 | 93.749 | 93.193 | 92.626 | 93.594 | 92.871 | 91.970 | 89.245 | 83.172 | 73.540 | 70.626 |
| Learned | × | 10 | 74.830 | 79.324 | 88.233 | 92.715 | 93.182 | 91.758 | 91.380 | 94.150 | 93.449 | 92.237 | 88.311 | 81.771 | 72.506 | 69.614 |
| **Translation on x-axis** | | | | | | | | | | | | | | | | |
| × | – | – | 33.556 | 44.144 | 61.628 | 78.501 | 87.543 | 89.323 | 89.456 | 89.345 | 88.166 | 80.080 | 59.426 | 38.805 | 29.619 | 28.262 |
| Fixed | ✓ | – | 93.582 | 94.595 | 95.262 | 95.863 | 95.907 | 95.918 | 95.718 | 96.018 | 95.974 | 95.974 | 95.596 | 94.862 | 93.827 | 92.270 |
| Fixed | × | 1 | 91.158 | 92.548 | 93.360 | 94.116 | 94.305 | 94.150 | 94.127 | 94.450 | 94.394 | 94.305 | 94.038 | 92.670 | 91.247 | 89.578 |
| Fixed | × | 3 | 91.859 | 92.014 | 92.048 | 92.526 | 93.160 | 92.615 | 93.204 | 93.638 | 93.071 | 92.626 | 91.603 | 92.070 | 92.159 | 91.580 |
| Fixed | × | 10 | 90.869 | 91.469 | 91.414 | 91.792 | 92.148 | 91.336 | 91.647 | 94.272 | 93.160 | 92.626 | 91.603 | 91.614 | 91.002 | 91.625 |
| Learned | ✓ | – | 86.498 | 86.865 | 92.159 | 95.462 | 96.029 | 95.707 | 95.618 | 95.762 | 95.685 | 94.684 | 93.004 | 89.634 | 83.628 | 83.684 |
| Learned | × | 1 | 76.543 | 79.391 | 88.377 | 93.004 | 93.627 | 92.492 | 92.181 | 94.150 | 93.616 | 92.370 | 89.434 | 82.894 | 74.063 | 71.716 |
| Learned | × | 3 | 74.964 | 78.167 | 87.543 | 92.726 | 93.749 | 93.193 | 92.626 | 93.594 | 92.871 | 91.970 | 89.245 | 83.172 | 73.540 | 70.626 |
| Learned | × | 10 | 74.830 | 79.324 | 88.233 | 92.715 | 93.182 | 91.758 | 91.380 | 94.150 | 93.449 | 92.237 | 88.311 | 81.771 | 72.506 | 69.614 |

we sample an independent random checkerboard for each image by uniformly assigning each cell in the $28 \times 28$ grid to either black or white. Rotations are applied to the digit mask using nearest-neighbor resampling with zero padding to preserve discrete pixel structure. For translations, we use circular shifts implemented via array rolling, thereby pixels shifted beyond the image boundary reappear on the opposite side rather than being clipped. The transformed digit mask is then overlaid onto the random background.

Upon loading, images are normalized to the range [0,1] and flattened from $3 \times 28 \times 28$ into a 2,352-dimensional feature vector. A linear encoder then maps this vector to a latent representation of dimension 70, which is also the dimensionality of the transformation operators. In the presence of transformations, the latent embedding is manipulated with the linear operator in the representation space. For compound transformations, the second encoder is a linear mapping of size $70 \times 70$ to adapt the canonical representation to the second transformation before canonicalizing.

All models are trained using the Adam optimizer with a learning rate of 0.001 and a batch size of 512 for 20 epochs and temperature $\lambda = 1$ in Eq.2. All experiments are conducted on a single NVIDIA RTX 5090 desktop GPU.

In the evaluation reported in Figure 2 the approriate operator was chosen by our k-NN methods. To that end, we shuffle the validation set using a fixed random seed (42), select 2,000 samples as the reference set, and perform nearest-neighbor classification with $k = 1$.

# C  ADDITIONAL RESULTS

## C.1  EXTRAPOLATION RESULTS

We report the absolute numerical accuracy for the extrapolation test visualized in Figure 2 in Table 1 and Table 2. The shaded columns indicate transformation degrees observed during training, while unshaded columns correspond to unseen (out-of-distribution) transformations.

We observe that the baseline model without operators experiences a severe accuracy degradation as the amount of distortion deviates substantially from the learned set. Although its performance is moderate within the observed region, accuracy drops sharply for larger unseen shifts and rotations, in some cases approaching near-random levels. This behavior indicates that the baseline fails to generalize beyond the transformation degrees encountered during training and must implicitly absorb transformation variability into its representation. For example, under vertical translation (along y-axis), baseline accuracy drops from 78.5–83.3% within the observed region to 13.6% at a shift of +10 and 15.2% at +12. Under rotation, the degradation is also pronounced: accuracy falls from

Table 2: **Classification accuracy (%) under rotation extrapolation.** "Degree given" indicates whether the true rotation degree is provided at test time. Blue-shaded columns denote degrees seen during training; unshaded columns are unseen.

| Operator | Degree given | k | Degree | | | | | | | | | |
|---|---|---|---|---|---|---|---|---|---|---|---|---|
| | | | -144 | -108 | -72 | -36 | 0 | 36 | 72 | 108 | 144 | 180 |
| × | – | – | 25.170 | 30.620 | 74.497 | 78.456 | 77.322 | 79.001 | 75.075 | 31.721 | 26.148 | 25.648 |
| Fixed | ✓ | – | 95.707 | 95.751 | 95.785 | 95.963 | 95.918 | 95.707 | 95.607 | 95.696 | 95.618 | 95.840 |
| Fixed | × | 1 | 86.042 | 85.841 | 86.753 | 86.965 | 86.842 | 86.976 | 86.720 | 85.975 | 85.875 | 86.564 |
| Fixed | × | 3 | 86.920 | 87.521 | 88.644 | 89.523 | 89.890 | 86.831 | 84.763 | 84.051 | 84.740 | 85.864 |
| Fixed | × | 10 | 87.298 | 87.443 | 87.721 | 88.099 | 88.511 | 89.834 | 88.377 | 87.821 | 87.843 | 87.810 |
| Learned | ✓ | – | 95.774 | 96.052 | 96.185 | 96.129 | 96.118 | 96.018 | 96.285 | 95.762 | 95.284 | 95.718 |
| Learned | × | 1 | 86.242 | 85.586 | 85.853 | 85.452 | 85.363 | 89.256 | 88.655 | 87.443 | 86.753 | 86.742 |
| Learned | × | 3 | 87.321 | 87.554 | 88.032 | 88.121 | 88.544 | 89.078 | 87.588 | 86.709 | 85.797 | 86.553 |
| Learned | × | 10 | 86.609 | 85.719 | 86.609 | 86.119 | 86.865 | 91.180 | 90.023 | 88.967 | 87.866 | 87.588 |

(a) Horizontal translation          (b) Vertical translation          (c) Rotation

Figure 5: **Extrapolation behavior of operator-based models on MNIST.** Solid lines use ground-truth transformation degrees; dashed lines "(auto)" use k-NN pose inference. Shaded regions denote training degrees.

78–79% in-domain to 25–31% at extreme angles (±144°, 180°). This sharp decline indicates poor extrapolation beyond the transformation degrees seen during training.

In contrast, operator-based models maintain stable performance across the full transformation range. When the true transformation is provided at test time ("Degree given" ✓), both fixed and learned operators achieve near-constant accuracy of 95–96% across all translation and rotation degrees, including unseen ones. Even when the transformation must be inferred ("Degree given" ×), performance remains high. For example, under rotation with automatic inference (k=1), accuracy remains around 85–87% across all angles—dramatically outperforming the baseline at extreme rotations. Similar stability is observed for translations, where operator models consistently remain above 90% even far outside the training range.

Figure 5 illustrates the accuracy behavior of operator-based models as a function of transformation magnitude. The results show that strong out-of-distribution performance is consistent for both pre-defined (blue) and learned (red) operators. The flat accuracy curves outside the shaded training region indicate that the operator formulation successfully generalizes beyond the observed transformation range. The similarity between pre-defined and learned operators suggests that the robustness stems from the operator-based factorization itself rather than a particular parameterization.

## C.2 ABLATION: HYPERPARAMETERS FOR K-NN SEARCH

We conduct an ablation to study the impact of reference set size $N$ and neighborhood size $k$ on k-NN search accuracy. For each $N \in 100, 200, 500, 1000, 2000, 5000$, we randomly sample a reference set and extract canonicalized embeddings using the optimized model. Test samples at each discrete transformation degree are classified via kNN search over the reference embeddings, with $k \leq N$. The results reported in Tables C.1, 4, and 5 are averaged across multiple random seeds $\{0, 10, 20, 30, 42\}$ and all possible transformation degrees.

Table 3: **k-NN performance on rotated MNIST.** Acc$_{pose}$ (↑) and Acc$_{cls}$ (↑) denote pose prediction and downstream classification accuracy, respectively. **GT** reports the reference performance when ground-truth poses were given. The highest score for each $N$ is shown in **bold**.

| $k$ | $N$ | | | | | | | | | | | |
|---|---|---|---|---|---|---|---|---|---|---|---|---|
| | 100 | | 200 | | 500 | | 1000 | | 2000 | | 5000 | |
| | Acc$_{pose}$ | Acc$_{cls}$ | Acc$_{pose}$ | Acc$_{cls}$ | Acc$_{pose}$ | Acc$_{cls}$ | Acc$_{pose}$ | Acc$_{cls}$ | Acc$_{pose}$ | Acc$_{cls}$ | Acc$_{pose}$ | Acc$_{cls}$ |
| **GT** | 100.000 | 95.759 | 100.000 | 95.759 | 100.000 | 95.759 | 100.000 | 95.759 | 100.000 | 95.759 | 100.000 | 95.759 |
| 1 | **54.196** | **76.136** | 60.339 | **80.185** | 65.292 | 83.375 | 68.912 | 85.473 | 71.543 | 87.029 | 74.159 | 88.726 |
| 3 | 51.829 | 73.990 | 59.669 | 79.097 | 66.285 | 83.136 | 70.421 | 85.512 | 73.273 | 87.171 | 75.949 | 88.940 |
| 10 | 53.287 | 75.149 | **61.767** | 80.054 | **68.687** | **84.356** | **72.950** | **86.649** | **75.746** | **88.117** | **78.735** | **89.771** |
| 30 | 49.344 | 72.227 | 58.294 | 77.425 | 66.211 | 82.657 | 71.258 | 85.480 | 74.676 | 87.288 | 78.143 | 89.172 |
| 100 | 40.792 | 66.761 | 50.160 | 72.366 | 59.804 | 78.436 | 65.642 | 82.134 | 70.056 | 84.612 | 74.985 | 87.326 |
| 300 | – | – | – | – | 51.437 | 73.311 | 58.320 | 77.275 | 63.375 | 80.187 | 69.657 | 84.169 |

Table 4: **k-NN performance on x-translated MNIST.** Acc$_{pose}$ (↑) and Acc$_{cls}$ (↑) denote pose prediction and downstream classification accuracy, respectively. **GT** reports the reference performance when ground-truth poses were given. The highest score for each $N$ is shown in **bold**.

| $k$ | $N$ | | | | | | | | | | | |
|---|---|---|---|---|---|---|---|---|---|---|---|---|
| | 100 | | 200 | | 500 | | 1000 | | 2000 | | 5000 | |
| | Acc$_{pose}$ | Acc$_{cls}$ | Acc$_{pose}$ | Acc$_{cls}$ | Acc$_{pose}$ | Acc$_{cls}$ | Acc$_{pose}$ | Acc$_{cls}$ | Acc$_{pose}$ | Acc$_{cls}$ | Acc$_{pose}$ | Acc$_{cls}$ |
| **GT** | 100.000 | 95.097 | 100.000 | 95.097 | 100.000 | 95.097 | 100.000 | 95.097 | 100.000 | 95.097 | 100.000 | 95.097 |
| 1 | **65.661** | **86.931** | **71.705** | **89.458** | 76.762 | **91.406** | 79.852 | **92.351** | 82.816 | **93.049** | 85.395 | **93.652** |
| 3 | 61.803 | 84.460 | 69.530 | 88.018 | 76.752 | 90.774 | 80.566 | 91.964 | 83.767 | 92.776 | 86.307 | 93.501 |
| 10 | 59.041 | 82.852 | 68.694 | 87.245 | **77.251** | 90.438 | **81.514** | 91.884 | **84.657** | 92.752 | **87.464** | 93.488 |
| 30 | 49.978 | 77.833 | 62.427 | 83.570 | 73.384 | 88.400 | 79.318 | 90.649 | 83.263 | 91.986 | 86.501 | 92.973 |
| 100 | 35.470 | 69.169 | 44.566 | 75.132 | 62.005 | 83.197 | 71.753 | 87.313 | 77.801 | 89.653 | 83.078 | 91.734 |
| 300 | – | – | – | – | 44.389 | 74.927 | 58.429 | 81.770 | 68.581 | 85.535 | 76.887 | 89.191 |

Table 5: **k-NN performance on y-translated MNIST.** Acc$_{pose}$ (↑) and Acc$_{cls}$ (↑) denote pose prediction and downstream classification accuracy, respectively. **GT** reports the reference performance when ground-truth poses were given. The highest score for each $N$ is shown in **bold**.

| $k$ | $N$ | | | | | | | | | | | |
|---|---|---|---|---|---|---|---|---|---|---|---|---|
| | 100 | | 200 | | 500 | | 1000 | | 2000 | | 5000 | |
| | Acc$_{pose}$ | Acc$_{cls}$ | Acc$_{pose}$ | Acc$_{cls}$ | Acc$_{pose}$ | Acc$_{cls}$ | Acc$_{pose}$ | Acc$_{cls}$ | Acc$_{pose}$ | Acc$_{cls}$ | Acc$_{pose}$ | Acc$_{cls}$ |
| **GT** | 100.000 | 95.913 | 100.000 | 95.913 | 100.000 | 95.913 | 100.000 | 95.913 | 100.000 | 95.913 | 100.000 | 95.913 |
| 1 | **66.732** | **87.442** | **72.056** | **90.092** | 77.052 | **92.161** | 79.872 | **93.020** | 82.513 | **93.792** | 85.342 | **94.532** |
| 3 | 63.211 | 84.416 | 71.428 | 88.416 | 77.668 | 91.441 | 80.967 | 92.725 | 83.756 | 93.651 | 86.227 | 94.502 |
| 10 | 62.334 | 84.603 | 71.036 | 88.648 | **78.322** | 91.380 | **82.019** | 92.753 | **84.673** | 93.655 | **87.319** | 94.521 |
| 30 | 57.436 | 83.192 | 66.505 | 86.956 | 76.085 | 90.423 | 80.460 | 91.934 | 83.504 | 93.067 | 86.603 | 94.138 |
| 100 | 47.862 | 78.718 | 55.208 | 82.766 | 67.933 | 87.820 | 75.309 | 90.169 | 80.057 | 91.618 | 83.930 | 93.080 |
| 300 | – | – | – | – | 56.196 | 83.258 | 64.537 | 86.892 | 73.189 | 89.365 | 79.839 | 91.525 |

**Effect of reference set size.** Across all transformations, both pose prediction accuracy (Acc$_{pose}$) and downstream MNIST classification accuracy (Acc$_{cls}$) improve consistently as the reference set size $N$ increases, since larger reference sets provide denser coverage of the canonical embedding space, and hence more reliable nearest-neighbor retrieval. However, performance gains begin to saturate between $N = 2000$ and $N = 5000$.

**Effect of neighborhood size $k$.** There appears to be a clear non-monotonic effect on performance in terms of $k$. The performance exhibits a clear non-monotonic dependence on $k$. Indeed, large neighborhoods ($k \geq 100$) dilute pose-specific information. Interestingly, even very small neighborhoods (e.g., $k = 1$) can often infer the correct pose, indicating that nearest matches in the embedding space are reliably informative. Across all transformations, a moderate value of $k$, most notably $k = 10$, consistently achieves the highest Acc$_{pose}$ and Acc$_{cls}$, potentially thanks to its robustness against noisy outliers and local pose structure.

**Comparison across transformations.** Despite the larger group order, translation-based tasks are generally easier than rotation, achieving higher pose and classification accuracy under identical

$(N, k)$ configurations. Nevertheless, we chose $N = 2000$ as the accuracy for finding the correct pose is typically around 70–80% and leads to negligible degradation in the downstream score.

**Using ground-truth transformations as labels.** We performed an ablation where the ground-truth transformation was provided at test time, and used as transformation label instead of the automatic k-NN pose inference label. As illustrated in Figure 5, automatic inference (dashed lines) yields slightly lower accuracy than the supervised inference with the ground-truth operator (solid lines), yet the performance gap remains modest across most transformation degrees. This result suggests that the model learns a well-structured latent space in which samples are correctly mapped to their canonical version through application of the latent operator, leading to robust pose estimation, disentanglement, and extrapolation.

**Future work.** Overall, predicted poses were not entirely correct, but the degradation remained small for several choices of hyperparameters. Notwithstanding this empirical success, our current pose prediction procedure does not scale well, as it relies on an exhaustive search over transformation candidates and distance comparisons against a selected reference set. This results in complexity that grows with both the number of transformation degrees and the size of the reference database. Future work could alleviate this limitation by developing more structured inference mechanisms—such as learning transformation-aware embeddings/classifiers or exploiting spectral decompositions—so that the pose can be inferred in linear time with respect to the embedding dimension rather than the transformation space.

