# OpenReview forum: "Latent Equivariant Operators for Robust Object Recognition: Promises and Challenges"
_ICLR.cc/2026/Workshop/GRaM — ICLR 2026 Workshop GRaM Poster_

### Official Review · Reviewer_7qEV · 2026-02-15
**Good for tiny paper**

**Rating:** 6
**Confidence:** 4

**Review:**

This paper studies latent equivariant operators for improving robustness to unseen transformations. Using a minimal MNIST setup, the authors show that learning a latent space operator enables stable extrapolation and composition of discrete rotations and translations, outperforming a non-operator baseline in out-of-distribution classification. While promising, the framework is evaluated only on invariant classification and still assumes known transformation structure during training.


Pros:

1, The paper provides a clean and controlled experimental setup that clearly demonstrates improved out-of-distribution generalization under unseen transformations.

2, The framework offers a flexible alternative to hardcoded equivariant architectures by learning latent operators instead of embedding explicit group structures into the network design.

Cons:

1, The claim in abstract that "....., but require knowledge of transformations a priori" is not clarified. The proposed method also assumes known transformation families and indices during training (e.g., discrete rotations/translations and their group structure), meaning it does not eliminate the need for prior transformation knowledge but rather shifts it from architectural design to data supervision. The authors should more clearly distinguish between requiring explicit architectural constraints and requiring labeled transformation information in training.

2, The proposed framework is evaluated only on invariant classification tasks, and it remains unclear how the method would extend to equivariant tasks (e.g., pose or rotation angle estimation), where the output should transform consistently with the input. Without experiments or analysis on equivariant downstream tasks, the generality of the learned operator beyond canonicalization for invariant classification is not established.

**Pmlr Suitability:**

NA

---

### Official Review · Reviewer_ikRE · 2026-02-24
**Interesting work suitable for tiny paper**

**Rating:** 6
**Confidence:** 4

**Review:**

## Summary
This study shows that latent symmetric operator methods can generalize classification performance across unobserved general transformations and combinations of general transformations using a minimal experimental design on rotated/translated noisy MNIST datasets. They compare three conditions i.e., no operator, predefined operators and learned operators, with concluding remarks discussing some of the difficulties associated with scaling their methods. Importantly, this project’s intent is to serve as a proof of concept rather than a new method.

## Strengths:
- The paper has an honest and well-scoped framing and within the body of work, the discussion of what the researchers found refegive strong empirical insights .

- The compound transformation result is the most compelling contribution. The fact that training on single axis transformations will only generalize to unseen compound transformations, without requiring O($N^2$) training pairs, is a significant and practical finding and indicates a leap forward from a pure reproduction of previous works. This is backed up by how well Figure 3 demonstrates this finding.

- The experimental design is clean and has a well-structured three-way comparison of the three different conditions (no operator, pre-defined, and learned) and resulted in unambiguous findings. The use of the heatmap in Figure 3 is a highly effective way to convey the generalization of the results on a single page.

## Weakness:

1. The novelty when compared to a Tiny Paper is low in both regards. The way that the core method is prescribed, is a direct adaptation of Bouchacourt et al (2021) who provide the operator construction without alteration. The proposed learned operator variant is also a reasonable extension but not a surprising one. There is little to no contribution apart from the empirical demonstration of the prior work, therefore there should be more explicit exposition on what is uniquely contributed, if anything beyond the previous work.

2. The dataset for the experiments is overly simplistic. MNIST with checkerboard background seemed like an appropriate basic test bed, however there is little semantic meaning in the checkerboard noise so it is much more likely to be ignored than the types of real world background variability that are present naturally. It is unclear if any of the results would generalize even for MNIST with natural backgrounds, let alone for real datasets. Therefore the type of results that are obtained in this study are very limit on the generalizability of them.

**Pmlr Suitability:**

NA

---

### Official Review · Reviewer_5ZLu · 2026-02-26
**Review for Latent Equivariant Operators for Robust Object Recognition**

**Rating:** 6
**Confidence:** 3

**Review:**

The paper demonstrates how equivariant latent operator methods can be successfully applied to an out-of-distribution classification problem.
The proposed approach generates two views of the given image by applying sampled transformations. The transformed views' representations are mapped to a canonical representation using the corresponding inverse shift operator, and an alignment loss between the canonicalized embeddings is applied along with the classification loss and a periodicity loss on the operator. During inference, the transformation label is extracted using kNN with a few prototypical samples with known transformations.  The test input embedding is evaluated under each candidate transformation operator, and the resulting embedding is further processed according to the majority-voted transformation.

My major concern is that the types of transformations used in the experiments may not encompass a wide range of transformations that are generally possible in real-world images. Further, the applicability of this idea to real-world examples is unknown without corresponding results.

**Pmlr Suitability:**

NA

---

### Meta-Review · Area_Chair_NKhB · 2026-02-26

**Decision:**

Accept

**Metareview:**

Reviewers agree that this paper is well aligned with the workshop themes and presents interesting contributions. Questions were raised about applicability to more complicated datasets. This could be addressed by the authors.

**Relevance To Proceedings:**

Tiny paper — does not apply

**Relevance To Workshop:**

Yes — suitable for GRaM

---

### Decision · Program_Chairs · 2026-03-02

Accept (Poster)